# Infigratinib (BGJ 398), a Pan-FGFR Inhibitor, Targets P-Glycoprotein and Increases Chemotherapeutic-Induced Mortality of Multidrug-Resistant Tumor Cells

**DOI:** 10.3390/biomedicines10030601

**Published:** 2022-03-03

**Authors:** Sergei Boichuk, Pavel Dunaev, Ilshat Mustafin, Shinjit Mani, Kirill Syuzov, Elena Valeeva, Firuza Bikinieva, Aigul Galembikova

**Affiliations:** 1Department of Pathology, Kazan State Medical University, 420012 Kazan, Russia; dunaevpavel@mail.ru (P.D.); shinjit.mani@gmail.com (S.M.); grop2019@gmail.com (K.S.); f.bikinieva@kazangmu.ru (F.B.); ailuk000@mail.ru (A.G.); 2Сentral Research Laboratory, Kazan State Medical University, 420012 Kazan, Russia; elena.valeeva@kazangmu.ru; 3Department of Radiotherapy and Radiology, Faculty of Surgery, Russian Medical Academy of Continuous Professional Education, 125993 Moscow, Russia; 4Department of Biochemistry, Kazan State Medical University, 420012 Kazan, Russia; ilshat.mustafin@kazangmu.ru

**Keywords:** ABC-transporters, paclitaxel, doxorubicin, FGFR-inhibitors, resistance, apoptosis, sensitization

## Abstract

The microtubule-targeting agents (MTAs) are well-known chemotherapeutic agents commonly used for therapy of a broad spectrum of human malignancies, exhibiting epithelial origin, including breast, lung, and prostate cancer. Despite the impressive response rates shortly after initiation of MTA-based therapy, the vast majority of human malignancies develop resistance to MTAs due to the different mechanisms. Here, we report that infigratinib (BGJ 398), a potent FGFR1-4 inhibitor, restores sensitivity of a broad spectrum of ABCB1-overexpressing cancer cells to certain chemotherapeutic agents, including paclitaxel (PTX) and doxorubicin (Dox). This was evidenced for the triple-negative breast cancer (TNBC), and gastrointestinal stromal tumor (GIST) cell lines, as well. Indeed, when MDR-overexpressing cancer cells were treated with a combination of BGJ 398 and PTX (or Dox), we observed a significant increase of apoptosis which was evidenced by an increased expression of cleaved forms of PARP, caspase-3, and increased numbers of Annexin V-positive cells, as well. Moreover, BGJ 398 used in combination with PTX significantly decreased the viability and proliferation of the resistant cancer cells. As expected, no apoptosis was found in ABCB1-overexpressing cancer cells treated with PTX, Dox, or BGJ 398 alone. Inhibition of FGFR-signaling by BGJ 398 was evidenced by the decreased expression of phosphorylated (i.e., activated) forms of FGFR and FRS-2, a well-known adaptor protein of FGFR signaling, and downstream signaling molecules (e.g., STAT-1, -3, and S6). In contrast, expression of MDR-related ABC-transporters did not change after BGJ 398 treatment, thereby suggesting an impaired function of MDR-related ABC-transporters. By using the fluorescent-labeled chemotherapeutic agent PTX-Alexa488 (Flutax-2) and doxorubicin, exhibiting an intrinsic fluorescence, we found that BGJ 398 substantially impairs their efflux from MDR-overexpressing TNBC cells. Moreover, the efflux of Calcein AM, a well-known substrate for ABCB1, was also significantly impaired in BGJ 398-treated cancer cells, thereby suggesting the ABCB1 as a novel molecular target for BGJ 398. Of note, PD 173074, a potent FGFR1 and VEGFR2 inhibitor failed to retain chemotherapeutic agents inside ABCB1-overexpressing cells. This was consistent with the inability of PD 173074 to sensitize Tx-R cancer cells to PTX and Dox. Collectively, we show here for the first time that BGJ 398 reverses the sensitivity of MDR-overexpressing cancer cells to certain chemotherapeutic agents due to inhibition of their efflux from cancer cells via ABCB1-mediated mechanism.

## 1. Introduction

Acquired resistance to the chemotherapeutic and targeted drugs remains the biggest challenge in clinical oncology. Despite the specific mechanisms involved in tumor resistance to the conventional and targeted-based therapies (e.g., secondary mutations of the targeted proteins), the malignancies might also acquire the universal mechanisms rendering them unsusceptible to the broad spectrum of anti-cancer agents. These mechanisms result in multidrug-resistance (MDR), which involves comprehensive and overlapping mechanisms, including (1) deregulation of apoptosis [1,2,3]; (2) epithelial-mesenchymal transition (EMT) [4,5,6]; (3) enhanced DNA damage repair (DDR) [7]; (4) overexpression of drug efflux pumps in cancer cells. This is characterized by an increased expression of ATP-binding cassette (ABC) transporters. Despite 48 members of the ABC transporter family being currently described, the clinical evidence of drug resistance was shown mainly for 3 members. This includes overexpression of the ABCB1, which is also known as multidrug resistance (MDR)-associated transporter P-glycoprotein (or MDR-1), ABCC1 (also known as multidrug resistance-associated protein 1 (MRP1)), and ABCG2 (also known as a breast cancer resistance protein (BCRP). These proteins are responsible for removing the drugs out of the cell to decrease their intracellular concentration, thus reducing the effectiveness of the chemotherapeutic agents. In particular, several groups of anti-cancer drugs are known as the substrates of ABCB1 and include vinca alkaloids, anthracyclines, epipodophyllotoxins, and taxanes [8]. Moreover, overexpression of ABCB1 in cancer cells can confer resistance to tyrosine kinase inhibitors (TKIs) such as imatinib and dasatinib [9,10]. Thus, overexpression of P-glycoprotein in cancer cells was shown as one of the well-known factors associated with poor therapeutic response in patients receiving cytotoxic and targeted-based anticancer therapies. ABCG2 also plays an important role in the development of MDR in cancer cells by providing the effective efflux of a broad spectrum of chemotherapeutic agents, including anthracyclines, methotrexate, mitoxantrone, topotecan, irinotecan, flavopiridol, and camptothecin-derived topoisomerase I inhibitors [11,12]. ABCC1, another member of the C subfamily of ABC transporters, has been shown to transport vinca alkaloids, various neutral and anionic hydrophobic compounds, and products of Phase II drug metabolism, including many glutathione and glucuronide conjugates [13,14,15,16]. Thus, targeting of ABC transporters was also suggested as a prospective therapeutic approach to overcome tumor resistance to the chemotherapeutic agents and targeted drugs [17,18,19,20]. However, most of the attempts to inhibit the function of ABC transporters lead to disappointing results for both the first (e.g., verapamil, cyclosporine A) and second (e.g., valspodar, biricodar, dexverapamil) generations of the ABC inhibitors, because of their unacceptable levels of toxicity and/or lower potential inhibiting effects. Despite second-generation inhibitors exhibited highest affinity and specificity for ABCB1 transporter, these inhibitors were also shown to be the substrates for other ABC transporters, such MRP-1, and BCRP, and drug-metabolizing enzymes such as CYP 3A4, thereby hampering their development due to the significant pharmacokinetic alterations and undesirable drug-drug interactions [21,22]. Thus, third-generation inhibitors were developed (e.g., tariquidar (XR9576), zosuquidar (LY335979), and elacridar (GF120918) and displayed the highest affinity to ABCB1 and fewer pharmacokinetic interactions [23]. Over the years several of the above-mentioned ABC inhibitors reached clinical evaluation, however, in the vast majority of cases, the results obtained were disappointing. For example, phase III trials with tariquidar were abandoned due to the toxicity [24]. The results of a phase III clinical trial with zosuquidar did not illustrate the benefits for the patients with acute myeloid leukemia [25]. Several clinical trials were prematurely terminated due to unexpected side effects, toxicity from adverse interaction with anticancer drugs, poor solubility, and limitations in the design of the studies (e.g., the patients were not selected based on tumor expression of ABC transporters, the absence of clinically validated imaging assays to detect ABC transporter function, etc.) [26,27]. As a result, no inhibitor of ABC transporters has been approved for clinical use so far. Therefore, it’s an urgent need for clinical oncology to find out the potent and effective inhibitors with minimal adverse effects.

Despite TKIs acting on the catalytic site of the tyrosine kinase domain by competing with ATP binding, which blocks the kinase activity, interferes with downstream signaling pathways, and subsequently impairs cell proliferation and survival, the multiple reports also illustrate that TKIs can also interfere with the function of ABC transporters. For example, nilotinib (Tasigna), a Bcr-Abl inhibitor approved for therapy of chronic myelogenous leukemia (CML) has been shown to inhibit the function of ABCB1-, ABCG2-, and ABCC10 transporters and reverses drug resistance [28,29]. Rivoceranib, a potent inhibitor of vascular endothelial growth factor receptor 2 (VEGFR-2), has been shown to reverse ABCB1- and ABCG2-mediated resistance in cancer cells [30,31]. Lapatinib (Tykerb), a dual tyrosine kinase inhibitor that interrupts the HER2/neu and epidermal growth factor receptor (EGFR) pathways, was approved for the use of breast cancer and other solid tumors, and was also shown to inhibit ABCB1- and ABCG2-mediated MDR in cancer cells [32].

Similarly, inhibition of FGF-signaling by TKIs was also shown to be an effective therapeutic approach for cancer patients. Indeed, the promising results of pemigatinib and infigratinib were shown in advanced unresectable cholangiocarcinoma harboring *FGFR2* fusions or rearrangements [33,34,35], and erdafitinib in metastatic urothelial carcinoma with *FGFR2* and *FGFR3* genetic aberrations [36]. Recent studies have shown the presence of overactivation of the FGFR-mediated pathway in the multiple TNBC cell lines and TNBC patient samples; this contributes to the disease progression, metastasis, and chemoresistance [37,38,39]. Also, recent findings pointed out the presence of an activated FGFR-mediated pathway in the HCC 1806 cell line resistant to a PARP inhibitor [40]. Thus, the anti-tumor activities of FGFR inhibitors might be considered as the prospective therapeutic option for TNBC therapies, especially in chemoresistant settings. This therapeutic option might be effectively realized due to the activity of FGFR inhibitors as potent inhibitors of ABC-transporters. However, limited data illustrating this possibility is currently available. For example, it was shown that PD 173074, FGFR1 inhibitor, reversed ABCB1-mediated multidrug resistance to colchicine (CH), paclitaxel (PTX), and vinblastine (Vin) in CH-resistant human epidermoid carcinoma cell subline and ABCB1-overexpressing HEK293 cells [41].

In this study, we examined whether infigratinib (BGJ 398), a potent and selective FGFR1-4 inhibitor can reverse the resistance of ABCB1-overexpressing cancer cells to the conventional chemotherapeutic agents PTX and doxorubicin (Dox), a topoisomerase II inhibitor. For this purpose, we used ABCB1-overexpressing PTX-resistant basal-like TNBC cell line HCC 1806—Tx-R, which was previously established in our laboratory [42]. Given that a broad spectrum of tumorous tissues acquires MDR phenotype due to the repeated exposure to chemotherapeutic agents, we also expanded our study to the cancer cell lines that originated from the epithelium of the gastrointestinal tract and bone tissues. Thus, we also included in our study ABCB1-overexpressing gastrointestinal stromal tumor (GIST T-1 Tx-R) cell subline, which was also generated in our laboratory, as shown elsewhere [43].

Here, we show for the first time that Infigratinib (BGJ 398), effectively re-sensitizes MDR-overexpressing cancer cells to certain chemotherapeutic agents and induces their apoptosis when used in combination with PTX or Dox. This effect of BGJ 398 was due to its ability to inhibit the efflux of these chemotherapeutic agents from cancer cells via the ABCB1-mediated mechanism.

## 2. Materials and Methods

### 2.1. Chemical Compounds

BGJ 398 and PD 173074 were purchased in SelleckChem (Houston, TX, USA). Paclitaxel (PTX) and doxorubicin (Dox) were obtained from Sigma-Aldrich (Merck KGaA, Darmstadt, Germany). Alexa488-conjugated PTX (Flutax-2) and was purchased in (Thermo Fisher Scientific, Inc., Waltham, MA, USA). Calcein acetoxymethyl ester (AM) was obtained from Abcam (Abcam, Cambridge, MA, USA). The chemicals were dissolved in dimethyl sulfoxide (DMSO) according to the manufacturer’s recommendations.

### 2.2. Cell Lines and Culture Conditions

The human basal-like TNBC cell line HCC 1806 was purchased from the American Type Culture Collection (Manassas, VA, USA). Paclitaxel (PTX)- resistant ABCB1-overexpressing TNBC subline (HCC 1806-Tx-R) was established in our laboratory, as shown elsewhere [42]. GIST T-1 was established from a metastatic pleural tumor from a stomach GIST and contained a heterozygous 57-base pair deletion (V570-Y578) in the KIT exon 11 [44]. PTX-resistant ABCB1-overexpressing GIST T-1R subline was also established in our laboratory via stepwise treatment with increasing concentrations of PTX, as shown elsewhere [43]. The cell lines indicated above were maintained in RPMI-1640 medium (Paneco, Moscow, Russia) supplemented with 15% fetal bovine serum (Gibco; Thermo Fisher Scientific, Inc., Waltham, MA, USA), 50 U/mL penicillin and 50 µg/mL streptomycin. The cell lines were cultured at 37 °C in a humidified atmosphere of 5% CO_2_ in an incubator (LamSystems, Miass, Russia).

### 2.3. Antibodies

The primary antibodies used for western blotting were anti-PARP (cat. no. 436400; Invitrogen; Thermo Fisher Scientific, Inc., Waltham, MA, USA), cleaved caspase-3 (cat. no. 9661S), phospho-NuMA Ser395 (cat. no. 3429), phospho-Histone H3 Ser10 (cat. no. 53348), phospho-FGFR Tyr 653/654 (cat. no. 3471), phospho- Tyr196 (cat. no. 3864), phospho-AKT S473 (cat. no. 4060P), phospho-p44/42 MAPK (Erk1/2) Thr202/Tyr204 (cat. no. 4370S), phospho-STAT-1 Ty701 (cat. no. 7649), phospho-STAT-3 Tyr705 (cat. no. 9145), phospho-GSK3B Ser9 (cat no. 5558) phospho-S6 Ribosomal Protein Ser235/236 (cat. no. 4858T) (Cell Signaling Technology Inc., Danvers, MA, USA), anti-ABC subfamily G member 2 (cat. no. sc-58222), MDR-1 (cat. no. sc-55510), MRP-1 (cat. no. sc-18835) (Santa Cruz Biotechnology, Dallas, TX, USA), and beta-actin (cat. No. A00730-200, GenScript, Piscataway, NJ, USA); HRP-conjugated secondary antibodies, anti-mouse immunoglobulin (Ig)G (cat. no. sc-2005) and anti-rabbit IgG (cat. no. sc-2004), were purchased from Santa Cruz Biotechnology, Dallas, TX, USA.

### 2.4. Western Blotting Analysis

To examine the protein expression in parental and Tx-R cells, whole-cell lysates (WCL) were prepared by scraping the cells growing as monolayer into radio-immunoprecipitation buffer (RIPA buffer) (25 mM Tris-HCl pH 7.6, 5 mM nEDTA, 150 mM NaCl, 0.1% SDS, 1% NP-40, 1% sodium deoxycholate) supplemented with the cocktail of protease and phosphatase inhibitors. The cellular lysates were further incubated for 1 h at 4 °C and clarified by centrifugation for 30 min at 11,400 rpm at 2 °C. The protein concentrations in WCL were calculated by the Bradford assay. The protein samples (20 µg) were loaded on the 4–12% Bis-Tris or 3–8% Tris-acetate NuPAGE gels (Invitrogen, Carlsbad, CA, USA) and upon completion of electrophoresis transferred to a nitrocellulose membrane (Bio-Rad, Hercules, CA, USA), membranes were probed with primary (1:1000 and incubated overnight at 4 °C), and secondary antibodies (1:1000 and incubated for 1 h at room temperature) and visualized by enhanced chemiluminescence (Western Lightning Plus-ECL reagent, Perkin Elmer, Waltham, MA, USA). The densitometry analysis of Western blotting images was performed by NIH ImageJ software, version 1.49 (Bethesda, MD, USA).

### 2.5. Crystal Violet Staining

Cells were seeded into p100 plates and treated with BGJ 398 (1 µM), PD 173074 (100 µM) and PTX (1 µM) alone or in combination for 96 h. Cell medium was aspirated and 5 mL of 0.5% crystal violet staining solution to each well was added, and incubated for 20 min at room temperature. Plates were washed 4 times in a stream of tap water and kept inverted on filter paper to air-dry 3 mL of 1% SDS was added in each plate and kept on a shaker at room temperature for 1 h. The optical density was measured at 570 nm wavelength using a MultiScan FC plate reader (Thermo Fisher Scientific, Waltham, MA, USA).

### 2.6. RNA Extraction and Reverse Transcription-Quantitative Polymerase Chain Reaction (RT-qPCR)

Parental and Tx-R cancer cells were seeded in p100 plates and treated with TKIs (BGJ 398 or PD 173074) and chemotherapeutic agents (PTX or Dox) alone or in combination for 48 h. TRIzol reagent (cat. no. BC032; Invitrogen; Thermo Fisher Scientific, Inc.) was used to extract total RNA, according to the manufacturer’s protocol, and re-suspended in diethyl pyrocarbonate-treated Н_2_О. RNA was reverse transcribed to cDNA using the Moloney murine leukemia virus reverse transcriptase kit (cat. no. SK021; Evrogen JSC, Moscow, Russia) following manufacturer’s protocol, and further subjected to quantitative-PCR. 1 µL template cDNA in total was used in the qPCR reaction, with 5x qPCRmix-HS SYBR (Evrogen JSC, Moscow, Russia) and forward and reverse primer 10 mM of each. The sequences of the primers and thermal cycling conditions used in this study are shown elsewhere [42]. qPCR was performed using the CFX96 Real-Time detection system (Bio-Rad Laboratories, Inc., Hercules, CA, USA). Assays for each experience sample were processed in parallel with reference gene *GAPDH* and the relative levels of each mRNA were normalized regarding *GAPDH*. The ∆∆Cq method was then used to calculate relative gene expression [45].

### 2.7. Cellular Survival Assay

Parental and Tx-R cancer cells (e.g., HCC 1806 and GIST-T-1) were plated on 96-well flat-bottomed plates (Corning Inc. Corning, NY, USA) and allowed to attach and grow for 24 h before treatment with BGJ 398, PD 173074, PTX or Dox that were introduced into the cell culture for 72 h with specified concentrations alone or in combination. Half-inhibitory concentrations (IC_50_) further named as IC_50_ values were defined as the concentration of the anti-cancer drug required to inhibit cellular growth by 50%. This data was normalized to the DMSO-treated control cells. To calculate the IC_50_ values of the chemotherapeutic agents indicated above, MTS reagent (Promega, Madison, WI, USA) was introduced into the cell culture for 1 h to assess the live cell numbers. The viability of the cells was assayed at 492 nm on a MultiScan FC plate reader (Thermo Fisher Scientific, Waltham, MA, USA). IC_50_ values were determined by using the IC_50_ Tool Kit (http://ic50.tk/, accessed on 10 January 2022).

### 2.8. Flow Cytometry

FACs analysis was used to examine the inhibition of P-gp-mediated transport from BGJ 398 or PD 173074-treated parental and Tx-R cancer cells. For this purpose, fluorescent compound Calcein AM was used. Briefly, cells were preincubated with one of FGFR inhibitors (BGJ 398—20 µM or PD 173074—50 µM) for 60 min. After that, cells were trypsinized, centrifuged, and were incubated with 100 nM Calcein AM (in phosphate-buffered saline) for 30 min at a temperature of 37 °C. After centrifugation, cells were incubated in culture medium in the presence of the FGFR inhibitors for 60 min at a temperature of 37 °C. Lastly, cells were washed with ice-cold phosphate-buffered saline and analyzed by flow cytometer BD FACSCanto II (Becton Dickinson Biosciences, Franklin Lakes, NJ, USA) by using BD FACSDiva Software, version 7.0.

To assess the intensity of Dox- or Flutax-2-mediated fluorescence, cells were preincubated with one of FGFR inhibitors (BGJ 398—20 µM or PD 173074—50 µM) for 60 min and further exposed to 40 µM of Dox or 3 µM of Flutax-2 (Alexa-488 labeled paclitaxel). After 60 min, chemotherapeutic agents were extensively washed out with FBS-free culture medium, cells were further cultured for 30 min in presence of FGFR inhibitors and analyzed. Cells were analyzed by flow cytometer BD FACSCanto II (Becton Dickinson Biosciences, Franklin Lakes, NJ, USA) by using BD FACSDiva Software.,version 7.0

The number of apoptotic cells were counted by using Muse Annexin V Et Dead Cell Kit (Merck KGaA, Darmstadt, Germany) according to the manufacturer’s instructions. Unstained and single-stained untreated cells were used as controls. Cells were analyzed in a Muse Cell Analyzer (Merck KGaA, Darmstadt, Germany).

For all the experiments indicated above at least 10,000 events were acquired for each sample. Results were presented as the percentage of desired cells relative to the total number of cells as mean ± standard deviation 4 biological repeats.

### 2.9. Immunofluorescence Staining

Cells were seeded on glass coverslips coated with poly-L-lysine (Sigma-Aldrich, St. Louis, MO, USA) and allowed to attach for 48 h before treatment. The cells were first incubated with one of FGFR inhibitors (BGJ 398—20 µM or PD 173074—50 µM) for 60 min and then incubated with 40 µM of Dox or 3 µM of Flutax-2 (Alexa-488 labeled paclitaxel) for an additional 60 min. After the wash-out with a pre-warmed FBS-free culture medium, the cells were additionally incubated in presence of FGFR inhibitors for 30 min. The non-fixed slides were counterstained with 3 µg/mL Hoechst 33342 (Thermo Fisher Scientific, Inc., Waltham, MA, USA) for 5 min to outline the nuclei and processed for fluorescence microscopy to obtain the merged images that were visualized on an Olympus BX63 fluorescence microscope (Olympus, Tokyo, Japan). Images were captured using a Spot advanced imaging system.

### 2.10. Molecular Docking

To identify the potential binding sites of BGJ 398 on the ABCB1, the molecular docking procedure was performed by using Schrodinger molecular modeling software, version 2021-2 (Schrödinger, Inc., New York, NY, USA, 2021). The Glide Docking XP mode was used to find the ABCB1-BGJ 398 supramolecular complex with minimal energy scoring function. The ABC1 structure (no. 6QEX) was downloaded from the Protein Data Bank (6QEX. Available online: https://www.rcsb.org/structure/6QEX, accessed on 28 February 2022). The protein structure was prepared before docking using the Protein Preparation Wizard module. Initial ligand structures were optimized using standard preparation procedures in the LigPrep module. Standard Induced FIT Docking Protocol is used to generate possible induced ABCВ1 conformations [46]. The representative conformation of the ligand-receptor complex was selected based on the maximum IDF score. The molecular mechanics/generalized Born surface area (MM-GBSA) method was used as a post-docking validation tool [47]. Based on the XP docked complex, we calculated the binding free energy (∆Gbind) and ligand strain energy of each drug.

### 2.11. Statistics

All the experiments were repeated a minimum of 3 times. The results are presented as the mean ± standard error (SE) for each group. Statistical analyses (Student’s *t*-test) were performed using Statistical software program version 7.0 (S.A. Glantz, McGraw Hill Education, NY, USA). *p* < 0.05 was considered to indicate a statistically significant difference. The degree of combination effects was quantified by using the R-package of the computational tool SynergyFinder (https://bioconductor.org/packages/release/bioc/html/synergyfinder.html). Highest single agent model (HSA) was used to calculate the synergy [48]. Means of normalized protein levels were compared by using the analysis of variance (ANOVA) with subsequent pairwise comparisons (Tukey HSD test) in R software, version 4.1.2.(R Foundation for Statistical Computing, Vienna, Austria; URL https://www.R-project.org/).

## 3. Results

### 3.1. Tx-R Cancer Cells Exhibit the Increased Expression of ABC-Transporters

Given that the tumor’s resistance to certain chemotherapeutic agents might be due to the overexpression of ABC-transporters regulating the efflux of chemotherapeutic drugs from cancer cells, we initially examined whether the expression of 3 major ABC-transporters (e.g., ABCB1, ABCG2 and ABCC1) was increased in PTX-resistant (Tx-R) cancer cells. For this purpose, we used Tx-R cancer sublines previously generated in our laboratory. The IC_50_ values for PTX, Dox and BGJ 398 in parental and Tx-R triple-negative breast cancer (TNBC) cell lines and gastrointestinal stromal tumor (GIST) cells are shown in Table 1. For example, we observed a substantial (~80-fold) increase of IC_50_ value for Dox in Tx-R HCC 1806 cells, when compared with naive HCC 1806. Similarly, ~25-fold increase of IC_50_ value for PTX was found in Tx-R HCC 1806 cells. This was in concordance with a significant increase of ABCB1 (i.e., P-glycoprotein) expression in all Tx-R cancer sublines when compared to parental cancer cells (Figure 1). Similar to ABCB1, the expression of ABCG2 was also increased in the НСС 1806 subline when compared with its parental counterpart (Figure 1A). This data was in concordance with an increased expression of *ABCB1* mRNA in both HCC 1806 and GIST Tx-R cell sublines (Appendix A). Moreover, the *MRP-1*, *2* and -*5* mRNAs levels were also significantly increased in HCC 1806-Tx-R (Appendix A). In contrast to the breast cancer cell lines, a moderate increase of *ABCB1* and *MRP-1* and -*5* was observed in Tx-R GIST cells when compared with their parental GIST T-1 cells (Appendix A). As expected, minor anti-proliferative and cytotoxic activities of BGJ 398 was observed for both parental and Tx-R sublines, thereby revealing an absence of activating mutations in the FGFRs in both HCC 1806 and GIST T-1 cells.

Thus, the resistance to PTX and Dox in TNBCs and GIST sublines indicated above might be due to the increased expression of ABC-transporters, in particular, ABCB1 protein.

### 3.2. PTX-Resistant Cancer Cells Exhibit an Increased Efflux of Chemotherapeutic Drugs

To corroborate these findings, we examined whether an increased expression of ABC-transporters in Tx-R cancer cell lines mediate an effective efflux of the chemotherapeutic agents and therefore decrease their sensitivity to PTX and Dox, as was shown in Table 1. For this purpose, naïve vs. resistant TNBCs and GIST were exposed to Alexa 488-labeled PTX (Flutax-2) for 1h and after PTX wash-out, the cells were cultured for an additional 1 h and subjected for immunofluorescence microscopy to examine the intracellular staining of Alexa-488 fluorescence dye. Data shown in Figure 2A, illustrates the increased intensity of Alexa-488 staining in the cytoplasm of cells after PTX treatment. This was observed for naïve (upper panel), but not in resistant HCC 1806 cells (lower panel), thereby revealing an excessive removal of Alexa-488-labeled PTX from Tx-R TNBCs. The similar differences were observed between the naïve vs. resistant TNBCs treated with Dox (Figure 2B). Moreover, an excessive removal of Dox was also observed in Tx-R GIST T-1 cells (Appendix A), thereby illustrating the accumulation of certain chemotherapeutic agents in Tx-R cancer cells exhibiting various tissue origin.

In agreement with immunofluorescence microscopy staining, FACs data illustrated an increase of fluorescence intensity (in FL2 channel) in Dox-treated cells (histograms shown in gray color), when compared to the basal fluorescence levels of the non-treated cells (histograms shown in green color) (Figure 3A). Strikingly, the dramatic decrease of the fluorescence intensity of Dox-mediated fluorescence (gray histograms) was found in HCC 1806 Tx-R cells (Figure 3A—right), when compared with the naive HCC 1806 cells (Figure 3A—left), thereby revealing the decreased intracellular concentration of Dox in Tx-R HCC 1806 cells.

Similarly, parental HCC 1806 cells exhibited an increased intensity of Alexa-488 staining when compared to the basal levels of the non-treated controls (Figure 3B—left), thereby revealing accumulation of Alexa-488-conjugated PTX in cancer cells. In contrast, minor differences in fluorescence intensity of Alexa-488-labeled PTX were observed between non-treated and PTX-treated HCC 1806 Tx-R cells (Figure 3B—right), thereby suggesting the removal of Alexa-488-labeled PTX from Tx-R HCC 1806 cells.

Taken together, this data illustrates the effective efflux of the chemotherapeutic agents (e.g., PTX and Dox) from Tx-R HCC 1806 breast cancer cell line overexpressing ABC-transporters.

### 3.3. BGJ 398 Enhances the Intracellular Drug Accumulation in Cancer Cells Overexpressing ABC-Transporters

To examine whether FGFR inhibitors inhibit the efflux of chemotherapeutic agents from Tx-R cancer cells, we performed FACs analysis of cancer cells treated with PTX or Dox alone and in presence of BGJ 398 (the potent and selective inhibitor of FGFR1-4). As was mentioned above, a significant decrease of the intensity Dox-mediated fluorescence was observed in Tx-R HCC 1806 when compared to the parental HCC 1806 cells, thereby revealing an increased efflux of Dox from the resistant TNBC cells (Figure 2B). Strikingly, BGJ 398 increased the intensity of Dox-mediated fluorescence in Tx-R cells (Figure 2B—right lower image), but not in naïve HCC 1806 cells (Figure 2B—right upper image), thereby illustrating an ability of this FGFR inhibitor to attenuate the function of ABC-transporters upregulated in the resistant TNBCs. Similar data were obtained by using the Flutax-2 (Alexa488-labeled PTX). Again, BGJ 398 increased the fluorescence intensity of Alexa488-labeled PTX in the resistant (Figure 2A—right lower image), but not in naïve HCC 1806 cells (Figure 3A—right upper image). Strikingly, an ability of BGJ 398 to inhibit the efflux of chemotherapeutic agents from cancer cells was not limited to Tx-R breast cancer cells and was also observed in the ABC-overexpressing Tx-R cancer subline exhibiting different tissue origin—e.g., Tx-R GIST T-1 cells (Appendix A), thereby suggesting the inhibition of ABC-transporters activity as a common mode of action of BGJ 398. Of note, PD 173074, a well-known FGFR1 and VEGFR2 inhibitor failed to increase intracellular concentrations of Dox in both (parental and Tx-R) HCC 1806 cancer cells, as shown in Appendix A. Similar data was obtained for these cancer cell lines treated with Alexa-488-labeled PTX in presence of PD 173074 (data is not shown). This was in consistency with immunofluorescence microscopy staining, illustrating inability of PD 173074 to increase Dox-induced fluorescence in Tx-R cells (e.g., HCC 1806 and GIST T-1) (Appendix A, respectively).

Collectively, our data illustrate that BGJ 398, a selective FGFR inhibitor, effectively modulates an efflux of PTX and Dox from Tx-R cancer cells overexpressing ABC-transporters, in particular, ABCB1.

### 3.4. BGJ 398 Attenuates the Activity of the ABCB1 Transporter in Cancer Cells

Next, we examined directly whether BGJ 398 down-regulates the activity P-glycoprotein in Tx-R resistant cancer cells. For this purpose, we used the Calcein AM, a well-known fluorescent ABCB1 substrate, and examined its expression in resistant breast cancer cells after BGJ 398 treatment. We found that BGJ 398 effectively inhibited the efflux of Сalcein from the resistant TNBCs (Figure 4B), thereby illustrating ABCB1 as a possible molecular target for this FGFR inhibitor. As expected, BGJ 398 have no impact on intracellular concentration of Сalcein in naive HCC 1806 cells (Figure 4A). This was in agreement with the absence of ABCB1 expression in parental HCC 1806 cells, as was shown in Figure 1A. In contrast, PD 173074, a well-known FGFR1 and VEGFR2 inhibitor do not have an impact on the efflux of Сalcein AM from the naive and resistant TNBC cells, as well (Figure 4A,B, respectively), thereby illustrating the novel (“off-target”) effect of BGJ 398 on ABCB1 transporter in TNBC cells.

### 3.5. BGJ 398 Restored PTX’s Ability to Deregulate Cell Cycle Progression in Tx-R Cancer Cells

Based on the data illustrating the intracellular accumulation of chemotherapeutic agents and fluorescent dyes in BGJ 398-treated Tx-R TNBCs, we examined whether the extended exposure of cancer cells to PTX will be responsible to deregulate cell cycle progression and restore PTX effect. For this purpose, we analyzed the expression of the proteins that are specifically accumulated in the M-phase. As expected, the expression of histone 3 phosphorylated at residue Ser 10 (pH3 Ser10) was significantly up-regulated after PTX treatment in naïve (Figure 5A), but not in Tx-R cells (Figure 5B), thereby revealing their resistance to PTX. Strikingly, BGJ 398 significantly increased expression of pH3Ser10 in Tx-R cells treated with PTX (Figure 5B), and this effect was not observed in naïve TNBCs. Similarly, Tx-R TNBCs treated with PTX in presence of BGJ 398 exhibited an increased expression of pNuma Ser395, a well-known marker of interphase. This data highlights the BGJ 398′s potency to restore PTX ability to deregulate the cell cycle progression of Tx-R cancer cells and induce their accumulation in the M-phase.

This was in agreement with the changes in the morphology of Tx-R HCC 1806 and GIST T-1 cells, which was evidenced by the accumulation of round-shape cells treated with a combination of PTX and BGJ 398 (Appendix A, respectively). Indeed, PTX used alone has not a significant impact on the confluence of the cell cultures and the numbers of round-shape cells in both Tx-cancer cell lines, whereas a significant increase in number of these cells was observed when this agent was used in combination with BGJ 398.

### 3.6. BGJ 398 Re-Sensitizes ABCB1-Overexpressing Cancer Cells to PTX and Dox

To examine this possibility, both naïve and Tx-R TNBCs were treated with BGJ398 and PTX alone, or in combination for 96 h and subjected for western blotting analysis to examine the expression of the well-known markers of apoptosis—the cleaved forms of PARP and caspase-3. As expected, no evidence of apoptosis was observed in both naïve and Tx-R HCC 1806 cells treated with FGFR inhibitor (Figure 6A,B, respectively), whereas PTX treatment resulted in a substantial increase of the cleaved PARP and caspase-3 in the naive, but not Tx-R HCC 1806 cells (Figure 6A,B, respectively), thereby revealing the resistance to PTX in Tx-R HCC 1806 subline. Strikingly, increased expression of the cleaved forms PARP and caspase-3 was found in Tx-R HCC1 806 cells treated with the combination of PTX and BGJ398 (Figure 6B,D), thereby illustrating BGJ398′s potency to reverse sensitivity to PTX in Tx-R TNBCs. Similar effect was found for HCC 1806 cells treated with Dox in presence of BGJ 398 (data is not shown). In consistency with these findings, Tx-R GIST T-1 cells also exhibited increased expression of apoptotic markers (e.g., cleaved forms of PARP and caspase-3) when treated with Dox (or PTX) in presence of BGJ 398 (Appendix A).

These findings were in consistency with FACs data, illustrating the significant increase of early apoptotic (i.e., Annexin V/PI-negative) cells at 48 h of post-treatment with the combination of PTX and BGJ 398 (Figure 7A,B). Similar data was obtained in cancer cells treated with Dox in presence of BGJ 398. In contrast, PD 173074 used in combination with PTX or Dox failed to increase apoptosis in Tx-R HCC 1806 cells (Appendix A). This was in agreement with our previous data illustrating low potency of this FGFR inhibitor to retain chemotherapeutic agents inside cancer cells (Appendix A).

An ability of BGJ 398 to sensitize cancer cells to PTX and Dox was also confirmed by the long-term viability assay and crystal violet staining. Neither BGJ 398 or PTX used alone inhibited proliferation and viability of HCC 1806-TxR cells, whereas the combination of these drugs provided significant anti-proliferative and cytotoxic effects (Figure 8A—middle panel). Densitometry analysis revealed a significant (~5-fold) decrease in cell growth in HCC 1806-TxR cells treated with the combination of BGJ 398 and PTX when compared to the cells treated with BGJ 398 or PTX alone (Figure 8B). Similar results were obtained when Tx-R HCC 1806 cells were treated with Dox in presence of BGJ 398 (Figure 8A—middle panel). Quantification analysis also revealed the decrease of viability of Tx-R HCC 1806 treated with combination of Dox and BGJ 398, whereas these drugs used alone have no effect on cellular viability (Figure 8B). Similar data were obtained in GIST T-1 Tx-R cells. Again, BGJ 398 effectively re-sensitized these cells to PTX and Dox, whereas chemotherapeutic agents used alone failed to reduce their viability (Appendix A), thereby revealing the MDR-associated phenotype in this particular cell subline. In contrast to BGJ 398, PD 173074 used alone or in combination with PTX or Dox have no inhibitory effects on the viability of Tx-R HCC 1806 cells (Figure 8A—bottom panel*,* and C). Of note, the anti-proliferative activities of BGJ 398 used in combination with PTX might be also due to the impairment of FGFR-signaling pathway, which was evidenced by the decrease of phosphorylated forms of FGFR1-2, adaptor protein FRS-2, and downstream signaling molecules, maintaining the cell survival and proliferation (STAT-1, 3 and S6) (Appendix A).

The ability of BGJ 398 to re-sensitize Tx-R HCC 1806 cells to the chemotherapeutic agents (e.g., PTX and Dox) was also evidenced by an MTS-based colorimetric assay and calculated by using the R-package computational tool of the Synergy Finder program. The values of synergy between these drugs are shown in Table 2 and illustrates a prominent synergism of BGJ 398 and PTX for both Tx-R cell lines (e.g., HCC 1806 and GIST T-1) (Figure 9). High synergy scores were also observed between BGJ 398 and Dox for both Tx-R cell lines indicated above (Table 2). In contrast, PD 173074 failed to sensitize Tx-R cell sublines to PTX and Dox, as well (Table 2).

Previous studies demonstrated that reversal of MDR phenotype in cancer cells might be due to the lowered expression of the ABCB1 by the tested compound and was not due to the direct inhibition of P-gp transporter [49,50]. WB and PCR data shown in Figure 10 and Appendix A, respectively, illustrated no evidence of decrease of ABCB1 expression in BGJ 398-treated cells in both transcriptional and protein levels. Similarly, expression of the other ABC-transporters did not change after BGJ 398 treatment (Appendix A), thereby suggesting about BGJ 398′s ability to sensitize MDR-overexpressing cancer cells to chemotherapeutic agents due to its interaction with this ABC transporter.

This might be due to the overloading of its drug-binding pocket (DBP) or inhibiting ATPase activity via binding to the ATP-binding site of ABCB1. To delineate between these possibilities, molecular docking studies were performed (see below).

### 3.7. Molecular Docking Analysis

To determine the possible binding site of BGJ 398 in ABCB1, we performed a molecular docking analysis. Given that BGJ 398 effectively inhibited the efflux of chemotherapeutic agents (Dox and PTX) from MDR-expressing cancer cells, we examined the drug-binding pocket (DBP) of ABCB1 as a potential binding site for this ligand. Taking into account that BGJ 398 inhibits FGFR-signaling due to its binding to ATP-binding pocket, we also performed the molecular docking of this inhibitor with NBD1 (nucleotide-binding domain 1) of ABCB1. Molecular docking analysis was composed of 2 steps. First, we performed standard docking by using Glide XP mode and further utilized the Induced Fit Docking mode. Finally, to assess the sensitivity to input ligand geometry of the resulting induced-fit receptor conformation, we performed the re-docking of minimized ligands using Glide XP mode. Next, we calculated the RMSD values between the ligand poses obtained by induced-fit docking and re-docking procedure. Tariquidar, a well-known agent exhibiting a high binding affinity to the DBP of ABCB1, was used for the internal control.

Induced Fit Docking is a compromise solution between standard docking procedure and the molecular dynamics [46]. By using the standard virtual docking studies, the ligands are docked into the binding site of a receptor where the receptor is held rigid and the ligand is free to move. However, protein molecules do not behave as the rigid structures, and can undergo side-chain or back-bone movements, or both, as a result of the ligand binding. The Induced Fit Docking protocol allows to predict the changes in the receptor conformation upon the ligand binding, thereby reducing the percentage of false-negative results and predicting more accurate protein-ligand conformations when compared with the standard docking procedure [51].

The molecular docking data demonstrated the drug-binding pocket (DBP) of ABCB1 as the most possible binding site for BGJ 398 (Figure 11).

This data was obtained in standard docking XP mode and induced docking protocol, as well (Table 3). The low RMSD values for BGJ 398 and Tariquidar indicate the stability and low levels of the fluctuations in the ligand poses in DBP of ABCB1. Also, the low ligand strain energy values of BGJ 398 and Tariquidar in DBP illustrate the minimal energy required for the adaptation to the receptor-bound conformation compared to NBD1 of ABCB1. Unexcitingly, the similar rates of ligand strain energy values were observed for the other FGFR inhibitor PD 173074. This was in contrast with its inability to retain PTX and Dox in Tx-R cancer cells, as was shown in the Appendix A. In addition, PD 173074 failed to re-sensitize Tx-R cancer cells to the chemotherapeutic agents PTX and Dox (Figure 8, Table 2). The possible explanations of these discrepancies are described in Discussion.

Collectively, we show here for the first time the off-target effect of BGJ 398 on ABCB1-overexpressing multi-drug resistant Tx-R cancer cells. This inhibitor of FGFR signaling effectively re-sensitized a broad spectrum of Tx-R cancer cells to certain chemotherapeutic agents (e.g., PTX and Dox) due to inhibition of their efflux from cancer cells. In contrast, PD 173074, the other inhibitor of FGFR1 signaling, failed to sensitize Tx-R cells to the Dox and PTX. This in turn correlated with inability of PD 173074 to retain PTX and Dox inside Tx-R cancer cells, therefore illustrating the novel off-target effect for BGJ 398 which is not associated with the inhibition of FGFR-signaling pathway.

## 4. Discussion

Despite significant progress in the development of novel chemotherapeutic agents exhibiting potent anti-tumor activities, the systemic adverse effects and rapid development of drug resistance remain the biggest challenges in current cancer therapies. Indeed, after initiation of anti-cancer therapy the tumors acquire resistance to the chemotherapeutic drugs via multiple mechanisms. These include activation of pathway-specific and/or non-specific mechanisms in tumor cells allowing them to escape from chemo- and targeted-based therapeutic agents. In particular, modification of the targeted enzyme or its downstream signaling molecules in tumors is considered as a major mechanism responsible for their acquired resistance to the tyrosine kinase inhibitors (TKIs) [52,53]. On the other side, DNA repair-deficient tumors that are initially responding well to the platinum compounds and/or poly (ADP-ribose) polymerase inhibitors develop resistance to these chemotherapeutic agents via restoration of the affected DNA repair pathway(s) [7]. In addition to the activation of tumor-specific pathways, cancer cells also develop non-specific mechanisms mediating their resistance to the DNA-damaging chemotherapeutic agents. This includes the activation of several biological modules, including an increased drug efflux resulting in multidrug resistance (MDR), inhibition of apoptosis, epithelial-mesenchymal transition, etc. The systemic and detailed analysis of the signaling pathways acquired tumor resistance to chemo- and targeted-based therapies are described in detail in the corresponding reviews [54,55]. 

Besides the well-known role of ABC-transporters in the efflux of the conventional anticancer agents from tumor cells, the emerging number of evidence demonstrates that MDR might also affect the anti-cancer activities of the tyrosine kinase inhibitors (TKIs). Indeed, ABCB1/Pgp/MDR1, ABCC1/MRP1, ABCC2/MRP2, and ABCG2/BCRP are involved in the development of resistance to TKIs. For example, ABCB1/P-gp/MDR1 was shown to facilitate the efflux of imatinib mesylate (IM) [56,57] and nilotinib [58] from cancer cells, thereby contributing to TKIs resistance. Similarly, imatinib [59,60], nilotinib [9], dasatinib [61], and danusertib [62] were found to be the substrates of ABCG2/BCRP transporter that increases the efflux of these TKIs from cancer cells. Finally, ABCC1/MRP1 was found as an effective transporter of imatinib from chronic myeloid leukemia (CML) cells [63], whereas sorafenib was proposed as a substrate for ABCC2/MRP2 [64], therefore causing sorafenib resistance of renal carcinoma cell lines in vitro. On the other hand, several TKIs have also been shown to inhibit the function of ABC-transporters, thereby interfering with drug efflux from cancer cells and providing an attractive therapeutic option to overcome MDR-related resistance to the conventional chemotherapeutic agents. For example, IM reversed the resistance to Dox in ABCB1-overexpressing cancer cells [65]. Similarly, IM restored the sensitivity of ABCG2-expressing cancer cells to topotecan via increasing the intracellular accumulation of this chemotherapeutic agent [66]. It was also reported that IM stimulated the ATP-ase activity in both ABCB1- and ABCG2-overexpressing cells, therefore indicating that RTKi acts as a substrate for both ABCB1 and ABCG2 [67]. Nilotinib, a second-generation of TKI, also inhibited the efflux function of ABCB1, ABCG2, and ABCC10 in cancer cells, thereby increasing the intracellular concentrations of the chemotherapeutic agents (e.g., PTX and doxorubicin) and potentiating their cytotoxic effects both in vitro and in vivo [28,68]. The ability of TKI’s to interfere with the function of ABC-transporters was also shown for a broad spectrum of the inhibitors, including dasatinib [69], ponatinib [70], gefitinib [71], erlotinib [72], lapatinib [32], sunitinib [73,74], etc.

Despite the multiple reports illustrating the ability of TKIs to interfere with the function of MDR-related ABC-transporters, limited data is currently available about the potential role of the inhibitors of FGF-signaling to impair the function of these proteins and their abilities to re-sensitize cancer cells to the conventional chemotherapeutic agents.

Indeed, Chen Z-S with colleagues demonstrated that PD 173074, FGFR1 inhibitor, reversed ABCB1-mediated multidrug resistance to colchicine, PTX, and vinblastine (Vin) in colchicine-resistant human epidermoid carcinoma cell subline and ABCB1-overexpressing HEK293 cells, as well. This effect was due to the decreased active efflux of [3H]-PTX in ABCB1-overexpressing cells observed in efflux assay. Importantly, PD 173074 did not have the effect of ABCB1 expression but stimulated the ATP-ase activity of ABCB1 in a concentration-dependent manner, indicating a direct interaction with this transporter [75]. These effects of PD 173074 were further reproduced by using the MRP7-overexpressing HEK293 cells, thereby illustrating its ability to interfere with ABCC10 [41]. Of note, multiple reports illustrate that besides P-glycoprotein, expression of ABCC10/MRP7 (unlike other MRPs) also contributes to the resistance of various cancer cells to taxanes [76,77] and can be utilized as a predictive biomarker for the resistance to PTX in several human malignancies, including a non-small cell lung cancer (NSCLC) [78]**,** breast cancer [79] Strikingly, it was also shown that MRP7 can affect the in vivo tissue sensitivity to taxanes independently from ABCB1 [80].

Taken together, these findings suggest that modulation of ABCB1 and ABCC10 activities by specific and reversal inhibitors may have a significant clinical value in management of a broad spectrum of human malignancies that are treated with taxanes, based on action in eliminating these drugs from tumor tissues.

To overcome the ABC-mediated MDR mechanisms the reversal MDR inhibitors were generated. As mentioned before, first generation of ABCB1/MDR1/P-gp inhibitors included the calcium channel blocker verapamil, the sodium channel blocker quinidine and the immunosuppressant cyclosporine A. Despite of several promising trials illustrating an improvement of overall survival (OS) rates and increased remission in patients with Pgp-positive myelodysplastic syndromes receiving intensive chemotherapy combined with the first line inhibitors indicated above [20,81,82], most of the trials with first generation ABC-inhibitors showed no clinical benefits of these blockers and/or were terminated due to the toxicity reasons [83,84]. Second generation P-gp inhibitors, such as valspodar or PSC-833, and biricodar or VX-710 were later generated to improve the potency over the first-generation inhibitors and decrease toxicity. Despite valspodar fulfilled the requirement for a higher-affinity non-toxic Pgp inhibitor, the results of the majority of trials were also disappointing and did not show significant clinical benefits [85,86,87,88]. Moreover, valspodar exhibited the unanticipated pharmacokinetic interactions by interacting with cytochrome P450 3A4 and interfering with drug metabolism. As an outcome, anti-cancer drug exposure was increased leading to the overdose of cancer patients, which in turn raised the incidence and severity of adverse effects of anti-cancer therapies due to the decreased anti-cancer drug elimination [89]. Therefore, third-generation of low-toxic and potent ABC-inhibitors exhibiting minimal pharmacokinetic interactions were developed. This include tariquidar (XR9576), laniquidar (R101933), zosuquidar (LY335979) and CBT-1. Indeed, Phase I clinical trials of tariquidar used in combination with paclitaxel or doxorubicin showed no significant side effects and pharmacokinetic interactions [90]. However, the Phase III clinical studies aimed to examine the effectiveness of tariquidar with first-line chemotherapy for patients with NSCLC were terminated due to the toxicity reasons, as was reviewed in [24]. Zosuquidar was found as a potent and selective P-gp inhibitor and demonstrated clinical efficiency when used in combination with chemotherapeutic agents for limited set of the patients with acute myeloid leukemia (AML) [91]. Despite the clinical trials aimed to examine the potency of zosuquidar to ehsnace effectiveness of anti-cancer drugs for patents with AML and myelodysplastic syndromes are currently ongoing, the other trials did not show no significant difference in progression-free survival (PFS) and OS for the patients with metastatic or recurrent breast cancer [92].

Thus, it is an urgent need to develop novel, potent, selective and well-tolerated ABC-inhibitors to enhance cytotoxicity of the chemotherapeutic agents during anti-cancer therapy. Alternatively, the activity of the well-known TKIs to inhibit the function of ABC transporters and therefore enhancing cytotoxic activities of anti-cancer agents have to be re-evaluated.

In the present study, we show for the first time that BGJ398, a potent and selective FGFR1-4 inhibitor, effectively sensitizes PTX-resistant TNBC cells to PTX. This was evidenced by a significant (~24-fold) decrease of resistance index in both Tx-R HCC 1806 cells treated with PTX in presence of BGJ 398 (Table 1). Similar effects were obtained for another Tx-R cell subline- Tx-R GIST T-1 cell line, as well (Table 1). This point was also supported by a dramatic increase of apoptotic markers (e.g., cleaved PARP and caspase-3, increased numbers of Annexin V-positive cells, etc.) in PTX-resistant TNBCs treated with the combination of PTX and BGJ 398 (Figure 6 and Figure 7, respectively) and was consistent with a high Synergy score (>10) observed for all Tx-R cancer cells treated with both compounds indicated above (Figure 9, Table 2). The aforementioned effects of BGJ 398 might be due to the impaired efflux of PTX from these cells, as was shown in Figure 2 and Figure 3. In addition to PTX, the efflux of Dox was also impaired from Tx-R TNBC cells treated with BGJ 398, thereby confirming the ability of this FGFR inhibitor to abrogate the MDR phenotype of TNBC cells. Of note, similar to PD 173074 [75], BGJ 398 did not have an impact on ABCB1 expression in both transcriptional and protein levels (Figure 10, Appendix A), thereby suggesting BGJ 398′s ability to inhibit the function of MDR-related ABC-transporters and, in particular, ABCB1. However, in contrast to the previous findings [41,75], PD 173074 failed to re-sensitize ABCB1-overexpressing HCC 1806 cancer cells to PTX and Dox (Figure 8, Appendix A, Table 2). This was consistent with low potency of PD 173074 to retain chemotherapeutic drugs (e.g., Dox) inside Tx-R breast cancer and GIST T-1 cells (Appendix A, respectively). In particular, this could be due to the low expression of *MRP7* in the Tx-R cancer sublines (as shown in Appendix A) used in current study. In addition, we did not find the differences in binding energy scores between BG 398 and PD 173074, as shown in Table 3, thereby illustrating both of these FGFR inhibitors might interact evenly with DBP of ABCB1. Despite the fact that the binding energy of PD 173074 with DBP of ABCB1 is comparable to the binding energy of BGJ 398, further experiments are needed to prove this fact and determine the stability of the PD 173074 pose over time using various unbiased/biased molecular dynamics methods.

Of note, an ability of BGJ 398 to re-sensitize PTX-resistant cancer cells to PTX was not limited to the TNBC Tx-R cell lines. The similar effect of BGJ 398 was observed for ABCB1-overexpressing gastrointestinal stromal tumor cells (GIST-T-1 Tx-R). Again, expression of apoptotic markers significantly increased when the cells indicated above were treated with combination of PTX and BGJ 398, when compared to the single treatment (Figure 6). Strikingly, we observed a prominent synergism for BGJ 398 and PTX for GIST Tx-R cells (Figure 9B), thereby illustrating high potency of BGJ 398 to restore sensitivity to PTX in Tx-R cancer cells exhibiting the different tissue origin. In contrast to BGJ 398, PD 173034 failed to re-sensitize GIST T-1 Tx-R cells to PTX (Figure 8 and Appendix A). This was in agreement with the lack of the synergism between PD 173034 and PTX/Dox in Tx-R cell lines (e.g., Table 2).

Given that Tx-R cancer sublines utilized in the present study exhibited high expression levels of ABCB1, we also examined BGJ 398′s ability to inhibit the activity of this ABC transporter. For this purpose, we utilized 2 fluorescent dyes known to be the substrates for ABCB1. For example, Calcein AM is known as a highly permeable MDR1 transporter substrate [93]. Similarly, Rhodamine 123, a member of the rhodamine family of fluorine dyes, also is commonly used to examine membrane transport by the ABCB1 gene product, MDR1 [94]. Indeed, our FACs data shown in Figure 4 illustrated that BGJ 398 significantly impaired the efflux of Calcein AM from Tx-R cancer cells, thereby providing the evidence of a molecular mechanism of BGJ 398-induced re-sensitization of TNBCs to PTX.

Given the BGJ 398 reversed sensitivity of ABCB1-overexpressing Tx-R cancer cells to PTX and Dox due to impaired efflux of these chemotherapeutic agents, we also examined its potential binding site in ABCB1. It was established that besides binding to membrane-bound and cytoplasmic tyrosine kinases, some of TKIs might also act as competitive inhibitors of ABC transporters by overloading the transporter’s drug-binding pocket (DBP) [95]. Alternatively, some TKIs (e.g., afatinib) bind with the ATP-binding site of ABC transporters to inhibit their ATP-ase activity [96,97]. In addition, some TKIs down-regulate expression of ABC-transporters [95].

To delineate between these possibilities, we initially compared the expression of MDR-related proteins between non-treated vs. BGJ 398-treated ABCB1-overexpressing cancer cells. As shown in Figure 10, BGJ 398 failed to decrease the expression of ABCB1 and other MDR-related proteins in both transcriptional and protein levels, therefore suggesting its ability to bind with ABCB1. The molecular docking data demonstrated the drug-binding pocket (DBP) of ABCB1 as the most possible binding site for BGJ 398 (Figure 11, Table 3). Indeed, despite BGJ 398 effectively binding to the ATP-binding pocket of FGFR2 to mediate its targeted activity, the affinity of this FGFR inhibitor to NBD of ABCB1 was significantly lower when compared with DBP (Table 3). This was evidenced by the data obtained by regular docking procedures and induced fit docking, as well. These differences might be due to two reasons. First, despite the functional similarities of ATP-binding sites of FGFR2 and ABCB1 (in both cases these sites mediate the ATP hydrolysis), the structures of ATP-binding sites in these molecules substantially differ from each other. For example, the ATP-binding site of NBD1 is phylogenetically more ancient and composed of highly conserved amino acid residues, also known as Walker A and Walker B motifs, as was discovered in 1982 [98]. In contrast, the ATP-binding site of tyrosine kinases domain is an evolutionally much younger protein-based domain that is mainly present in eukaryotic cells [99]. This domain is also composed of highly conserved motifs, such as Glycine-rich loop, HDR, DGF, etc. that substantially differ from Walker A and Walker B in ABCB1 (Appendix A). Therefore, the significant differences in the structures of ATP-binding sites between ABCB1 and FGFR2 might explain the low binding scores observed for BGJ 398 in the ATP-binding site of NBD1. Second, DBP of ABCB1 interacts with the chemotherapeutic agents via several flexible alpha spirals, directed to the top of the inward-facing conformation [100,101]. Based on such flexibility of the alpha spirals in ABCB1, binding of BGJ 398 with a flexible transmembrane site is much more probable, when compared with the stable and highly conserved ATP-binding site of NBD1.

This data is consistent with the previous findings that illustrated the possibility to inhibit ABCB1 activity by the RTKis. Indeed, besides BGJ 398, IM was also shown to bind with the transmembrane domain, but not with NBD of ABCB1 [67]. Similar activities were recently found for FGFR1 inhibitor, PD 173074 [41]. The RTKis indicated above significantly increased ATP-ase activity of ABCB1, thereby indicating the ability to bind with a drug-binding pocket of ABCB1.

Taken together, this data suggests the interaction of BGJ 398 with the amino acids of a drug-binding pocket of ABCB1 and is consistent with recent findings indicated above. Further experiments by using advanced methods (e.g., unbiased/biased molecular dynamics, FEP calculation, X-ray diffraction, or Cryo-EM) will be helpful to illustrate the precise pose of the BGJ 398 and its potential interactions with the transmembrane domain of ABCB1.

Collectively, we show here that BGJ 398, a selective FGFR1-4 inhibitor, inhibits the efflux function of the ABCB1 transporter and effectively re-sensitizes MDR-expressing cancer cells to certain chemotherapeutic agents, including PTX and Dox. Our current data is in close consistency with recent findings of Wu S with co-authors, illustrating a high potency of erdafitinib, a small-molecule pan-FGFR kinase inhibitor to induce sensitization of ABCB1-overexpressing cancer cell lines to certain chemotherapeutic drugs, including PTX, vincristine, and topotecan [102]. It was also found that erdafitinib stimulates ATPase activity in ABCB1-expressing cells, thereby indicating that this RTKi acts like a substrate for ABCB1. Moreover, study from Kim SH et al. showed that PTX used in combination with BGJ 398 synergistically suppressed urothelial carcinoma cell migration and colony formation via regulation of EMT-associated factors, while FGFR1 knockdown enhanced the antitumor effect of PTX [103]. This in turn suggests that BGJ 398 might be considered as a potent RTKi to enhance cytotoxic activities of anti-cancer agents for ABCB1-overexpresing human malignancies, whereas high levels of ABCB1 in tumor tissues could be further evaluated as a biomarker for treatment selection of BGJ 398 and PTX.

Taking into account that acquired resistance to the conventional chemotherapies in TNBC is commonly associated with the development of MDR phenotype after initiation of the chemotherapy, our data illustrate the BGJ 398 (infigratinib) as a potential therapeutic option for MDR-related TNBCs, especially for the subset of patients with TNBCs harboring FGFR alterations. Indeed, activation of FGFR-signaling pathway in breast cancer was evidenced by the multiple studies [104,105], and analyzed in detail in several recent reviews to illustrate the role of FGFR signaling in breast cancer pathogenesis and progression [106] and provide the molecular basis for ongoing clinical trials [ClinicalTrials.gov Identifier: NCT02365597]. Moreover, our recent data demonstrated activation of FGFR-signaling in GIST might be an alternative mechanism of secondary resistance to IM [107] and therefore serve as an attractive molecular target to re-sensitize GIST to IM both in vitro and in vivo [108]. Besides the BGJ 398-induced impairment of ABCB1 function shown here, we recently demonstrated an ability of this RTKi to sensitize cancer cells to topoisomerase II inhibitors (e.g., doxorubicin) via attenuation of homology-mediated DNA repair [109], thereby illustrating the complex mechanism of BGJ 398-induced sensitization of cancer cells to the chemotherapeutic agents.

## Figures and Tables

**Figure 1 biomedicines-10-00601-f001:**
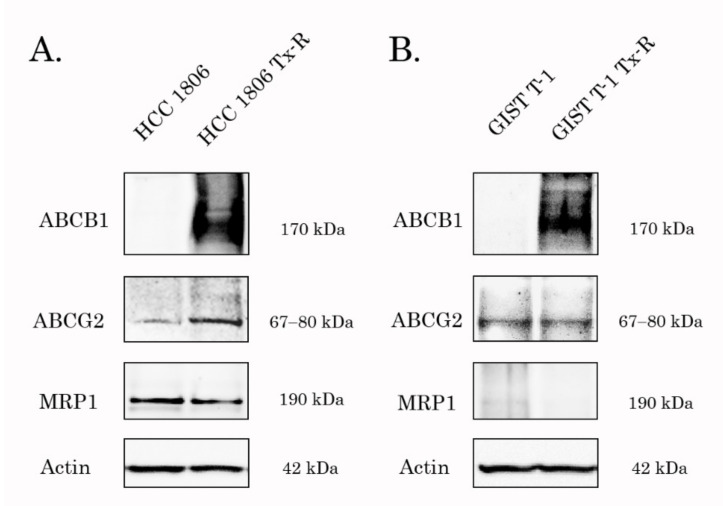
Expression of ABCB1, ABCC1 (MRP-1), and ABCG2 transporters in Tx-R cancer cells vs. Tx-sensitive parental HCC 1806 (**A**) and GIST T-1 (**B**) cells. Actin stain was used as a loading control.

**Figure 2 biomedicines-10-00601-f002:**
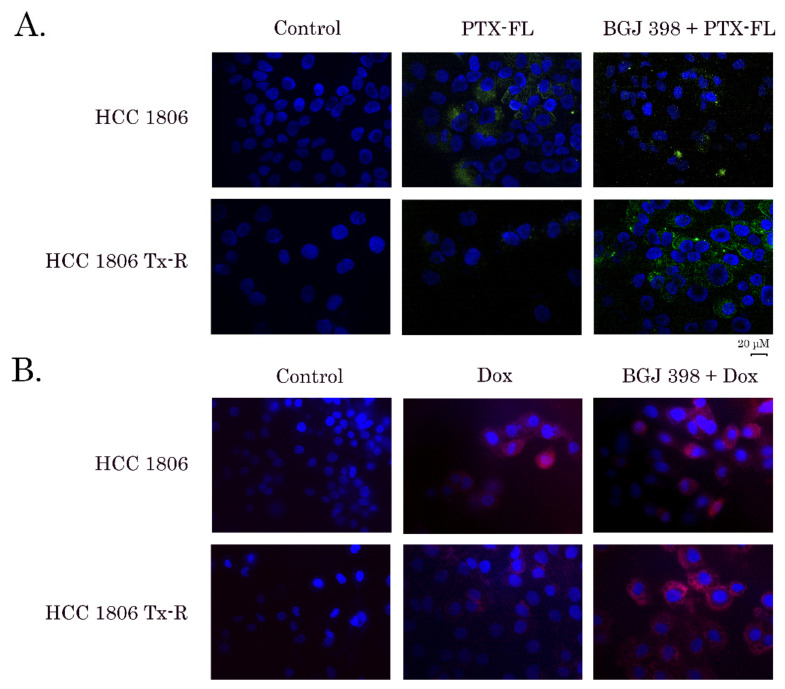
Fluorescence microscopy analysis of the intracellular accumulation of Flutax-2 (PTX-FL) (**A**) and doxorubicin (DOX) (**B**) in HCC 1806 parental (upper panels) and Tx-R (lower panels) cells. The cells were first incubated with BGJ 398 (20 µM) (right) or DMSO as a control (middle) for 60min and then incubated with 3 µM PTX-FL (**A**) or 40 µM DOX (**B**) for an additional 60 min. After the wash-out with a pre-warmed FBS-free culture medium, the BGJ 398-treated cells were additionally incubated with BGJ 398 for 60 min. The non-fixed slides were counterstained with Hoechst 33342 (final concentration 3 µg/mL) for 5 min to outline the nuclei and processed for fluorescence microscopy to obtain the merged images.

**Figure 3 biomedicines-10-00601-f003:**
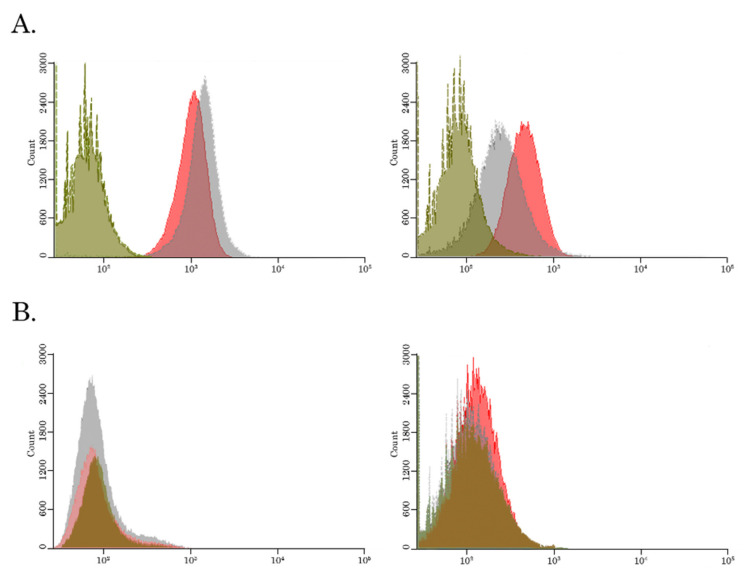
BGJ 398 increases the intracellular accumulation of Dox (**A**) and Flutax-2 (**B**) in Tx-R HCC 1806 breast cancer cells. (**A**) The intracellular accumulation of Dox in drug-sensitive parental HCC 1806 cells (left) and Tx-R HCC 1806 cells (right). The cells were treated with DMSO (green), Dox (40 µM) alone (gray) or in combination with BGJ 398 (20 µM) (red). (**B**) The intracellular accumulation of Alexa-488 labeled PTX (Flutax-2) in drug-sensitive parental HCC 1806 cells (left) and Tx-R HCC 1806 cells (right). The cells were treated with DMSO (green), Flutax-2 (3 µM) alone (gray) or in combination with BGJ 398 (20 µM) (red). The fluorescence intensity was analyzed by FACs. Representative histograms of at least 3 independent experiments are shown.

**Figure 4 biomedicines-10-00601-f004:**
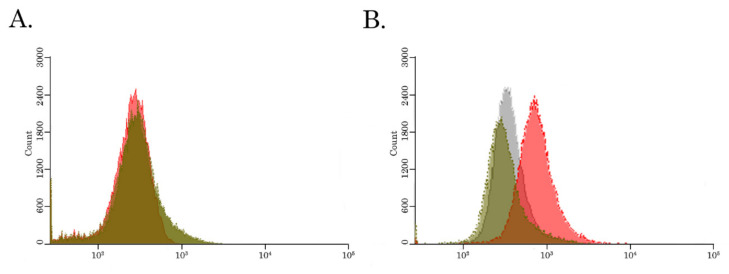
BGJ 398 increases intracellular accumulation of Calcein, an ABCB1 fluorescent substrate, in ABCB1-overexpressing Tx-R HCC 1806 cells. The intracellular accumulation of Calcein, a fluorescent product of the ABCB1 substrate Calcein AM, was determined in the Tx-R HCC 1806 cells (**B**) and the corresponding drug-sensitive parental HCC 1806 cells (**A**). The cells indicated above were treated with 100 nM of Calcein AM alone (green) or in combination with 20 µM of BGJ 398 (red) or 50 µM PD 173074 (gray). The fluorescence intensity was analyzed by FACs. Representative hist grams of at least 3 independent experiments are shown.

**Figure 5 biomedicines-10-00601-f005:**
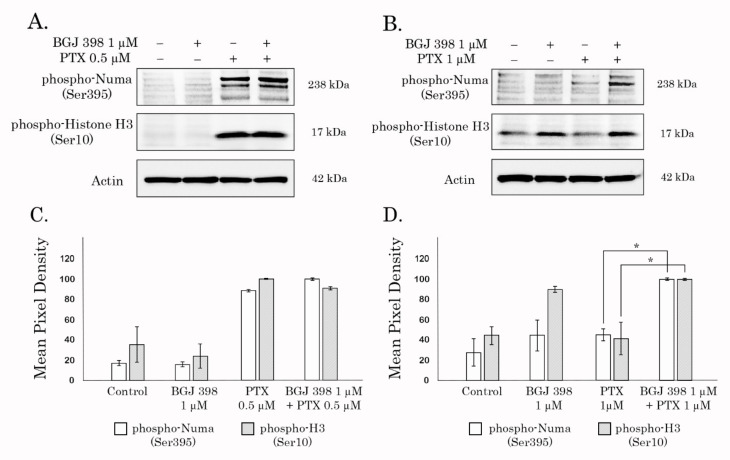
BGJ 398 potentiates accumulation of Tx-R HCC 1806 cancer cells in M-phase after the treatment with PTX. Immunoblot analysis for phospho-NuMA (Ser395) and phospho-H3 (Ser10) in the parental (**A**) and Tx-R (**B**) HCC 1806 breast cancer cells after treatment with DMSO (negative control), BGJ 398, PTX alone or in combination for 48 h. Actin stain is used as a loading control. (**C**,**D**) Quantification by mean pixel density of pNuMA (Ser395) and pH3 (Ser10) in the parental (**C**) and Tx-R (**D**) HCC 1806 breast cancer cells. Values are means ± SD, *n* = 3. * *p* < 0.05 vs. untreated cells.

**Figure 6 biomedicines-10-00601-f006:**
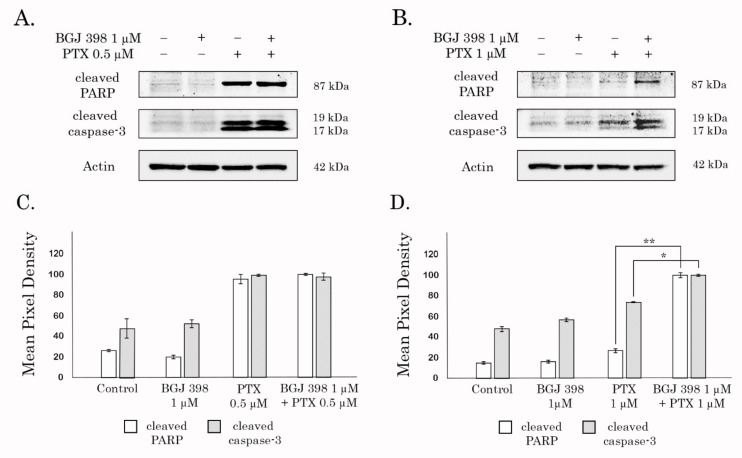
BGJ 398 potentiates apoptosis of Tx-R HCC 1806 cells treated with combination of PTX. Immunoblot analysis for apoptosis markers (cleaved forms of PARP and caspase-3) in naive (**A**) and Tx-R (**B**) HCC1 806 breast cancer cells after treatment with DMSO (negative control), PTX, BGJ 398 alone or in combination for 48 h. Actin stain is used as a loading control. (**C**,**D**) Quantification by mean pixel density in the cleaved forms of PARP and caspase-3 in naive (**C**) and Tx-R (**D**) HCC 1806 breast cancer cells. Values are means ± SD, *n* = 3. * *p* < 0.05, ** *p* < 0.01.

**Figure 7 biomedicines-10-00601-f007:**
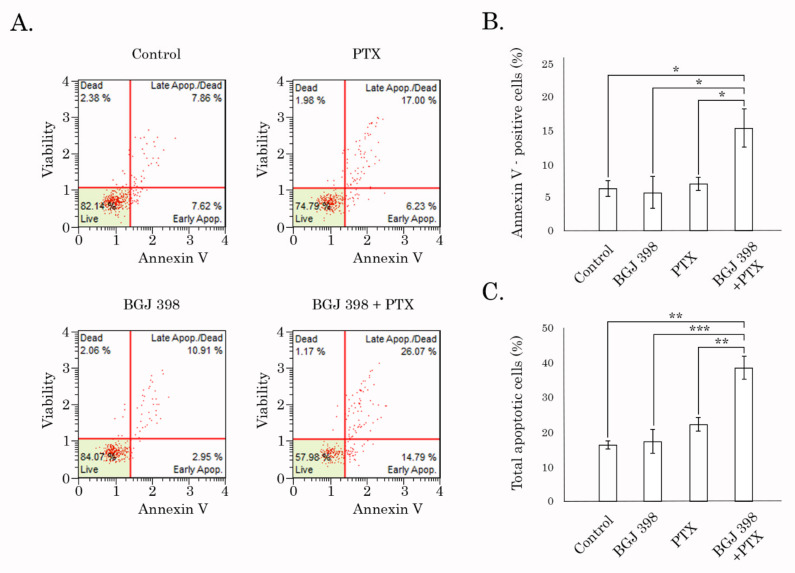
FACs analysis for apoptotic markers in Tx-R HCC 1806 breast cancer cells treated with DMSO (negative control), PTX (1 µM), BGJ 398 (1 µM) alone or in combination for 48 h. (**A**) Representative dot-plot are shown. (**B**) Quantitative analysis of the early apoptotic cells after the treatment as indicated above. (**C**) Quantitative analysis of the total apoptotic cells after the treatment as indicated above. * *p* < 0.05, ** *p* < 0.01, *** *p* < 0.001.

**Figure 8 biomedicines-10-00601-f008:**
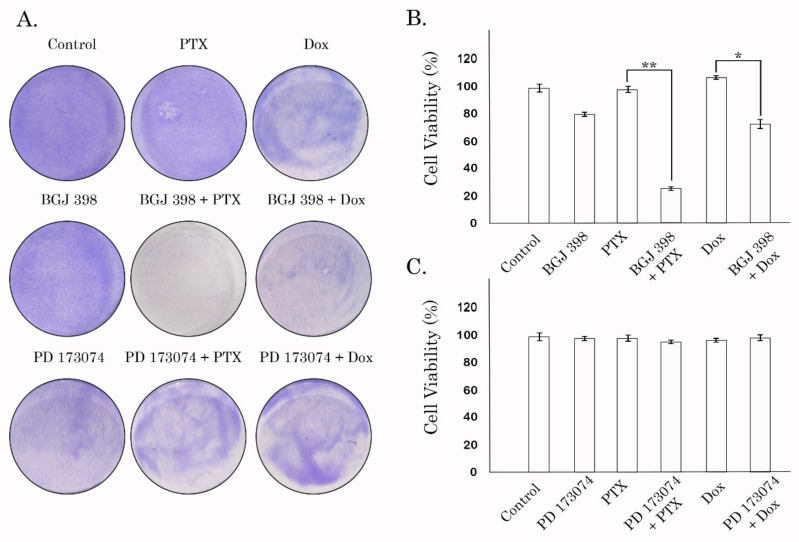
The impact of FGFR inhibitors (BGJ 398 and PD 173074) used in combination with PTX or Dox on proliferation and survival of Tx-R HCC 1806 breast cancer cells; (**A**) Crystal violet staining of Tx-R HCC 1806 cells that were treated with BGJ 398 (or PD 173074) alone or in combination with PTX (or Dox) for 96 h. The cells treated with DMSO were used as a control. The cells treated with PTX or Dox alone illustrated chemoresistance of Tx-R HCC 1806 cells. The culture dishes were fixed with ice-cold 100% methanol, stained with crystal violet, and photographed; (**B**) Quantification of crystal violet staining of Tx-R HCC 1806 cells treated with PTX and Dox alone or in combination with BGJ 398. The plates were dried, crystal violet was dissolved using 0.1% SDS solution, and absorbance was measured at 590 nm. The graphs represent the mean ± SD. * *p* < 0.05; ** *p* < 0.01; (**C**) Quantification of crystal violet staining of Tx-R HCC 1806 cells treated with PTX and Dox alone or in combination with PD 173074.

**Figure 9 biomedicines-10-00601-f009:**
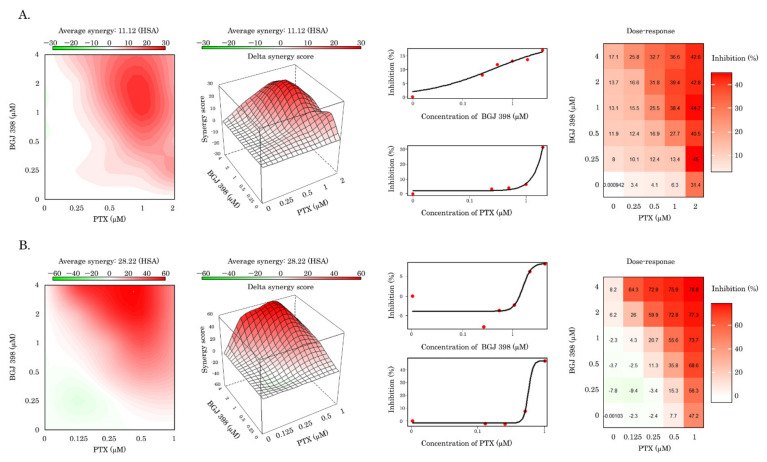
Assessment of the synergy between PTX and BGJ 398 in Tx-R HCC 1806 breast cancer (**A**) and GIST T-1 (**B**) cells. The average synergy for Tx-R HCC 1806 cells was 11.12, for GIST T-1 was 28.22.

**Figure 10 biomedicines-10-00601-f010:**
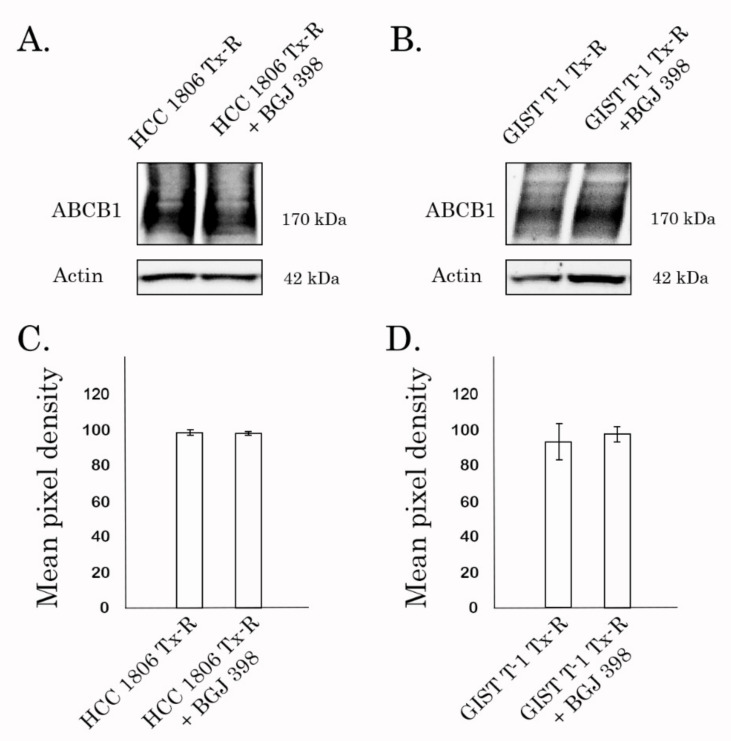
Expression of ABCB1 in Tx-R HCC 1806 (**A**) and GIST T-1 (**B**) cancer cells treated with DMSO (control) and BGJ398 for 72 h. Actin stain was used as a loading control. Actin stain was used as a loading control. (**C**,**D**) Quantification by mean pixel density of ABCB1 in Tx-R HCC 1806 cells (**C**) and GIST T-1 (**D**) cancer cells after treatment as indicated above. Values are means ± SD, *n* = 3. Statistically significant differences were not observed.

**Figure 11 biomedicines-10-00601-f011:**
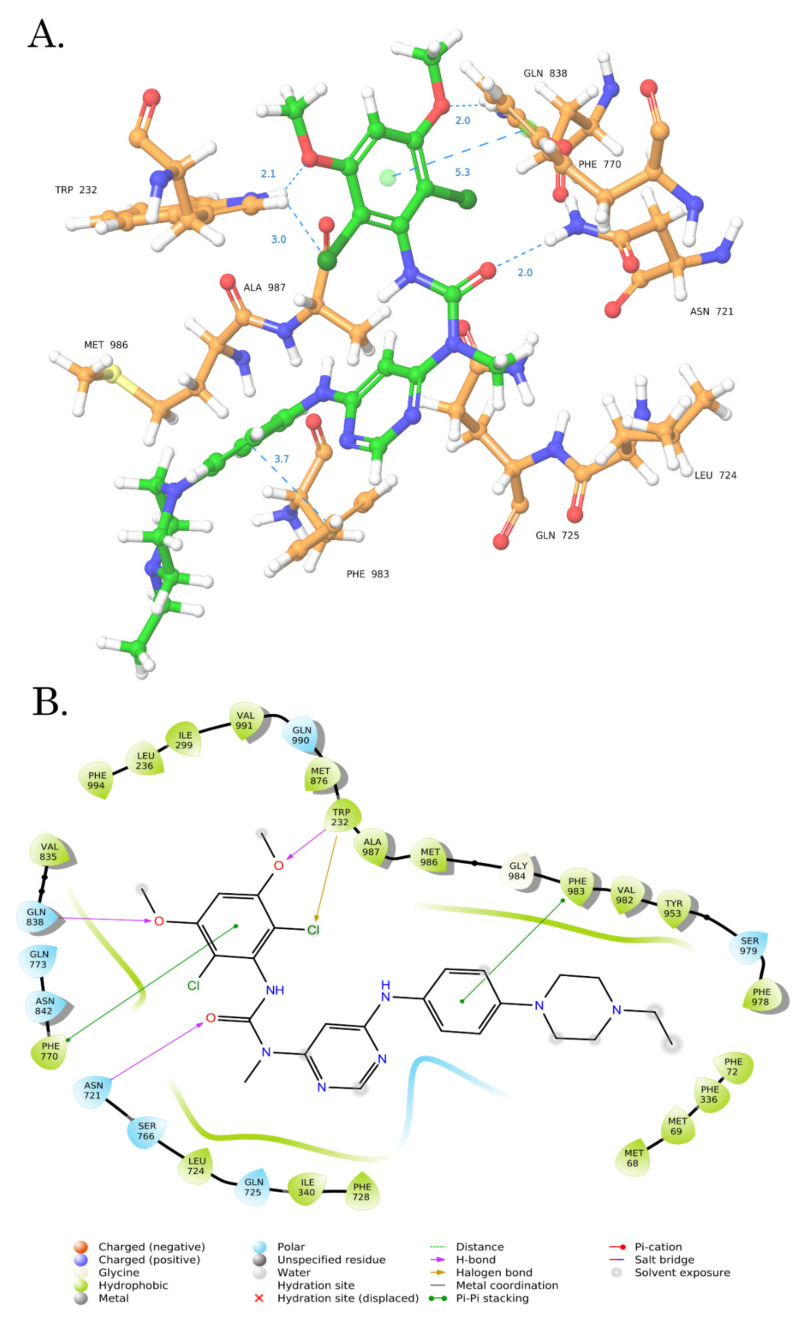
Computer modeling of BGJ 398 binding pose on DBP of ABCB1. (**A**) 3D diagram illustrating the proposed binding mode of the interactions of BGJ 398 with the ABCB1. (**B**) 2D Ligand interaction diagrams of the BGJ 398 with significant amino acid residues.

**Table 1 biomedicines-10-00601-t001:** IC_50_ values for PTX and Dox in parental and Tx-R triple-negative breast cancer (TNBC) and gastrointestinal stromal tumor (GIST) cell lines.

Chemical Compounds	Cell Line	Parental	Tx-R	Fold Increase
PTX (µM)	HCC 1806	0.22 ± 0.01	5.4 ± 1	24.5
	GIST T-1	<0.01	0.46 ± 0.08	>46
Dox (µM)	HCC 1806	0.22 ± 0.004	1.8 ± 0.5	81.8
	GIST T-1	0.04 ± 0.005	6.5 ± 1	162.5
BGJ 398 (µM)	HCC 1806	12 ± 1.3	17.6 ± 3	1.47
	GIST T-1	8.8 ± 1	7.2 ± 0.2	0.82

**Table 2 biomedicines-10-00601-t002:** Synergy scores between FGFR inhibitors and chemotherapeutic agents in Tx-R HCC 1806 and GIST T-1 cell lines.

Chemical Compounds	HCC 1806 Tx-R	GIST T-1 Tx-R
BGJ 398 + PTX	11.12	28.22
BGJ 398 + Dox	10.26	21.88
PD 173074 + PTX	0.22	7.27
PD 173074 + Dox	2.8	2.71

**Table 3 biomedicines-10-00601-t003:** Docking score (SP and XP Modes), MM-GBSA ∆Gbind prime energy, Ligand strain energy, RMSD of BGJ 398, Tariqiudar and PD 173074 with DBP and NBD1 of ABCB1.

DBP of ABCB1						
BGJ398						
	Docking Type	Standart Docking	Induced Fit Docking	RMSD	MM-GBSA ΔG_bind_ (kcal/mol)	Ligand Strain Energy (kcal/mol)
Scoring Function						
Glide XP score		−5.866	−9.605	1.74 Å	−84.36	4.8
Tariquidar						
Glide XP score		−7.249	−10.100	3.2 Å	−113.0	14.5
PD 173074						
Glide XP score		−8.968	−10.617	2.7 Å	−101.2	6.2
NBD1 of ABCB1						
BGJ398						
	**Docking Type**	**Standart Docking**	**Induced Fit Docking**	**RMSD**	**MM-GBSA** **ΔG_bind_** **(kcal/mol)**	**Ligand Strain Energy** **(kcal/mol)**
**Scoring** **Function**						
Glide XP score		−3.276	−4.459	12.7 Å	−58.7	10.4
Tariquidar						
Glide XP score		−5.752	−7.568	7.9 Å	−79.4	18.6
PD 173074						
Glide XP score		−5.147	−6.257	5.2 Å	−63.10	6.7

## Data Availability

Not available.

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
