# Peer review of "Infigratinib (BGJ 398), a Pan-FGFR Inhibitor, Targets P-Glycoprotein and Increases Chemotherapeutic-Induced Mortality of Multidrug-Resistant Tumor Cells"

_biomedicines, 2022, doi:10.3390/biomedicines10030601_

Round 1

Reviewer 1 Report

The manuscript from Dr. Sergei Boichuk and group identified a novel mechanism of action of Infigratinib (BGJ 398). BGJ 398, a pan-FGFR inhibitor, targets P-glycoprotein (ABCB1) and increases chemotherapeutic-induced mortality of multidrug-resistant (MDR) tumor cells. BGJ 398 blocks the function of ABCB1 as opposed to altering the ABCB1 protein expression in cells. Further, BGJ 398 increases the intracellular accumulation of paclitaxel by impairing its efflux from the triple-negative breast cancer (TNBC) and gastrointestinal stromal tumor (GIST) cell lines that are resistant to paclitaxel. A significant increase of apoptosis was evidenced by an increased expression of cleaved forms of PARP, caspase-3, and increased numbers of Annexin V-positive cells. Moreover, BGJ 398 used in combination with paclitaxel significantly decreased the viability and proliferation of the resistant cancer cells. Furthermore, inhibition of FGFR-signaling by BGJ 398 was evidenced by the decreased expression of phosphorylated (i.e. activated) forms of FGFR and FRS-2, a well-known adaptor protein of FGFR signaling, and downstream signaling molecules (e.g. STAT-1, -3, and S6). 

Overall, the manuscript by Dr. Boichuk and group is scientifically sound. This study addresses the issue of MDR in cancer, which have been studied by several research groups. This study identified previously unknown role of BGJ 398 in MDR reversal in TNBC and GIST cell lines resistant to paclitaxel.

Study from Kim SH et al. showed that paclitaxel+BGJ 398 synergistically suppressed urothelial carcinoma cell migration and colony formation via regulation of EMT-associated factors, while FGFR1 knockdown enhanced the antitumor effect of paclitaxel. Please note to include this important study that addresses reversal of paclitaxel resistance in your manuscript under "Discussion" section. Citation of Kim SH study can be found here:

Kim SH, Ryu H, Ock CY, Suh KJ, Lee JY, Kim JW, Lee JO, Kim JW, Kim YJ, Lee KW, Bang SM, Kim JH, Lee JS, Ahn JB, Kim KJ, Rha SY. BGJ398, A Pan-FGFR Inhibitor, Overcomes Paclitaxel Resistance in Urothelial Carcinoma with FGFR1 Overexpression. Int J Mol Sci. 2018 Oct 15;19(10):3164. doi: 10.3390/ijms19103164. PMID: 30326563; PMCID: PMC6214101.

Since ABCC10 (MRP7) is also an MDR pump that confers resistance to paclitaxel, was there any effort done to study the effect of BGJ 398 on ABCC10-overexpressing cells? It is noted that the authors cited the work of another similar FGFR inhibitor, PD173074, that reverses ABCC10-mediated multidrug resistance. It is suggested to include a discussion related to ABCC10-mediated paclitaxel resistance as such and studies done to reverse that. This additional information would give a complete picture of the paclitaxel resistance mediated by ABC transporters. Also, there have been some reports of some resistance to paclitaxel by other MRP pumps; it is recommended to include that in this study.

Historically there has been limited utility of MDR reversal agents clinically, which might be the case with BGJ 398. Please include the discussion related to clinical limitations of MDR reversal agents and how BGJ 398 might be a superior agent to advance in clinics as an MDR reversal agent. Also, please discuss if paclitaxel resistance mediated by ABCB1 to BGJ 398 could be used as a biomarker for treatment selection.

Including these aforementioned points in the discussion would increase the interest of the reader and at the same time provide useful content in regards to BGJ 398 reversal of MDR in cancer cells.

Author Response

We thank a reviewer for the comments and suggestions regarding our manuscript. Below are our specific responses to these comments (shown in quotes and italics). The changes in the revised version of the manuscript are highlighted with yellow.

“The manuscript from Dr. Sergei Boichuk and group identified a novel mechanism of action of Infigratinib (BGJ 398). BGJ 398, a pan-FGFR inhibitor, targets P-glycoprotein (ABCB1) and increases chemotherapeutic-induced mortality of multidrug-resistant (MDR) tumor cells. BGJ 398 blocks the function of ABCB1 as opposed to altering the ABCB1 protein expression in cells. Further, BGJ 398 increases the intracellular accumulation of paclitaxel by impairing its efflux from the triple-negative breast cancer (TNBC) and gastrointestinal stromal tumor (GIST) cell lines that are resistant to paclitaxel. A significant increase of apoptosis was evidenced by an increased expression of cleaved forms of PARP, caspase-3, and increased numbers of Annexin V-positive cells. Moreover, BGJ 398 used in combination with paclitaxel significantly decreased the viability and proliferation of the resistant cancer cells. Furthermore, inhibition of FGFR-signaling by BGJ 398 was evidenced by the decreased expression of phosphorylated (i.e. activated) forms of FGFR and FRS-2, a well-known adaptor protein of FGFR signaling, and downstream signaling molecules (e.g. STAT-1, -3, and S6). Overall, the manuscript by Dr. Boichuk and group is scientifically sound. This study addresses the issue of MDR in cancer, which have been studied by several research groups. This study identified previously unknown role of BGJ 398 in MDR reversal in TNBC and GIST cell lines resistant to paclitaxel.”

“Study from Kim SH et al. showed that paclitaxel+BGJ 398 synergistically suppressed urothelial carcinoma cell migration and colony formation via regulation of EMT-associated factors, while FGFR1 knockdown enhanced the antitumor effect of paclitaxel. Please note to include this important study that addresses reversal of paclitaxel resistance in your manuscript under "Discussion" section. Citation of Kim SH study can be found here:

Kim SH, Ryu H, Ock CY, Suh KJ, Lee JY, Kim JW, Lee JO, Kim JW, Kim YJ, Lee KW, Bang SM, Kim JH, Lee JS, Ahn JB, Kim KJ, Rha SY. BGJ398, A Pan-FGFR Inhibitor, Overcomes Paclitaxel Resistance in Urothelial Carcinoma with FGFR1 Overexpression. Int J Mol Sci. 2018 Oct 15;19(10):3164. doi: 10.3390/ijms19103164. PMID: 30326563; PMCID: PMC6214101.”

We appreciate a reviewer for these comments and suggestions and included the manuscript illustrating a high potency of BGJ 398 to sensitize urothelial carcinoma cells for paclitaxel via regulation of EMT-associated factors in the reference list (reference #103). This issue was also addressed in Discussion (lines 806-809).  

Since ABCC10 (MRP7) is also an MDR pump that confers resistance to paclitaxel, was there any effort done to study the effect of BGJ 398 on ABCC10-overexpressing cells?”  We are greatly appreciating the reviewer for this comment and suggestion. We did not examine the effect of BGJ 398 on ABCC10-overexpressing cancer cells so far and will consider these experiments as a part of our future plans”. 

“It is noted that the authors cited the work of another similar FGFR inhibitor, PD173074, that reverses ABCC10-mediated multidrug resistance. It is suggested to include a discussion related to ABCC10-mediated paclitaxel resistance as such and studies done to reverse that. This additional information would give a complete picture of the paclitaxel resistance mediated by ABC transporters. Also, there have been some reports of some resistance to paclitaxel by other MRP pumps; it is recommended to include that in this study”. We appreciate the reviewer for these comments and suggestions and expanded the discussion illustrating the role of MRP pumps in resistance to paclitaxel. The new data illustrating a high potency of ABCC10 and the other MRP pumps in cancer resistance to paclitaxel was addressed in the text highlighted with yellow (lines 655-665).

“Historically there has been limited utility of MDR reversal agents clinically, which might be the case with BGJ 398. Please include the discussion related to clinical limitations of MDR reversal agents and how BGJ 398 might be a superior agent to advance in clinics as an MDR reversal agent. Also, please discuss if paclitaxel resistance mediated by ABCB1 to BGJ 398 could be used as a biomarker for treatment selection”. We agree with the reviewer for these comments and suggestions and included the discussion about clinical limitations of MDR reversal agents. We also mentioned about the perspectives to use BGJ 398 as an MDR reversal agent and ABCB1 as a potential marker for treatment selection (lines 667-701, 809-812).  

We are now submitting our revised manuscript for your kind consideration of publication in Biomedicines.

Reviewer 2 Report

In this study, the authors identified that infigratinib can inhibits ABCB1 transporter, thereby antagonize multidrug resistance when combined with paclitaxel and doxorubicin. There are several issues need to be addressed.

  1. It is very hard to track the references because the citation and Reference list is not matching. Please revised the format.
  2. It is necessary to determine the cytotoxicity curve of BGJ 398 in sensitive and drug-resistant cancer cells. Why did the authors select 20 μM of BGJ 398 for the experiments?
  3.  In Figure 2, I suggest the author looks for another representive image to replace those of HCC 1806 Tx-R cells. The green fluorescence of PTX is too weak to see a significant difference.
  4. I wonder if BGJ is a substrate of ABCB1 transporter? What is the mechanism of its ABCB1 efflux inhibition?
  5. The authors may compare the binding score of BGJ 398 and PD 173074 to see if there are any difference.

Author Response

We thank the reviewer for the detailed analysis of our manuscript and comments and suggestions. Below are our specific responses to the reviewer’s comments (shown in quotes and italics). The changes in the revised version of the manuscript are highlighted in yellow.

Below are some suggestions that would be good for the readers, and can improve the article quality.

 1) “In this study, the authors identified that infigratinib can inhibits ABCB1 transporter, thereby antagonize multidrug resistance when combined with paclitaxel and doxorubicin. There are several issues need to be addressed. It is very hard to track the references because the citation and Reference list is not matching. Please revised the format”.

We greatly appreciate the reviewer for these comments and suggestions. We added the new citations and revised the format. The citations match with Reference list in the revised version of the manuscript. 

2) “It is necessary to determine the cytotoxicity curve of BGJ 398 in sensitive and drug-resistant cancer cells. Why did the authors select 20 μM of BGJ 398 for the experiments?”

We appreciate the reviewer for these comments and suggestions and included the cytotoxicity assay data (IC50) for BGJ 398 in sensitive and resistant cancer cell lines in the revised version of the manuscript. This data is currently shown in Table I.  Regarding the reviewer’s question about the concentration of BGJ 398 used for our studies. For the vast majority of assays that required the long-term culturing of the cells we used 1 mM of BGJ 398.  20 μM of BGJ 398 was only used for the short-term (1h) experiments (shown in Figures 2 and 3) to examine the impact of FGFR inhibitor on intracellular concentrations of chemotherapeutic agents. The point to use such concentration of BGJ 398 was due to the high levels of chemotherapeutic agents that are commonly used for this assay. In particular, concentration of Dox was 40 mM. This concentration of Dox was taken according to the multiple reports and protocols (e.g. Zhi Dai, et.al. MMP2-Sensitive PEG–Lipid Copolymers: A New Type of Tumor-Targeted P-Glycoprotein Inhibitor. ACS Appl. Mater. Interfaces 2016, 8, 20, 12661–12673.  https://doi.org/10.1021/acsami.6b03064). To examine an impact of the efflux of such amounts of Dox higher concentrations of RTKi are generally used. For example, according to the literature data imatinib mesylate was used at concentration of 40 mM to impair an efflux of Dox used at 10 mM (e.g. Sims JT., et al. Imatinib Reverses Doxorubicin Resistance by Affecting Activation of STAT3-Dependent NF-κB and HSP27/p38/AKT Pathways and by Inhibiting ABCB1. PLoS One. 2013; 8(1): e55509. doi: 10.1371/journal.pone.0055509). That’s why we used 20 mM of BGJ 398 for these experiments.  

3) “In Figure 2, I suggest the author looks for another representative image to replace those of HCC 1806 Tx-R cells. The green fluorescence of PTX is too weak to see a significant difference”.

We appreciate the reviewer for this comment and suggestion. We agree with a reviewer that intensity of green fluorescence was too weak in HCC 1806 Tx-R cells. To improve the quality of this Figure, we replaced this image for the new one to make the differences in the signal intensity more convincing.  

4) “I wonder if BGJ is a substrate of ABCB1 transporter? What is the mechanism of its ABCB1 efflux inhibition?”.

 We greatly appreciate a reviewer for these questions. The experiments to examine whether BGJ 398 can be a substrate of ABCB1 transporter are currently ongoing. In particular, we are doing FACs analysis by using the specific Abs (UIC2 and 5D3) for the “shift assay” and also measuring ATP-ase activity of P-gp by using the Pgp-Glo™ Assay.  These experiments are not finalized yet, so we did not include this data into this manuscript.

5)The authors may compare the binding score of BGJ 398 and PD 173074 to see if there are any difference. We appreciate a reviewer for this comment and suggestion and also included the binding score of PD 173074 in Table III. This data was also explained in Results and Discussion sections (lines 582-585 and 726-730).  

We appreciate the reviewer for all the comments and suggestions and submitting the revised manuscript for your kind consideration of publication in Biomedicines.

Round 2

Reviewer 2 Report

The authors have successfully addressed all my concerns.